# Association between predialysis creatinine and mortality in acute kidney injury patients requiring dialysis

Hsin-Hsiung Chang[1,2,3]*, Chia-Lin Wu[4,5], Chun-Chieh Tsai[4], Ping-Fang Chiu[4,6,7]

1 Division of Nephrology, Department of Internal Medicine, Antai Medical Care Corporation Antai Tian-Sheng Memorial Hospital, Dongguan, Taiwan, 2 Division of Nephrology, Department of Internal Medicine, Paochien Hospital, Pingtung, Taiwan, 3 Department of Computer Science and Information Engineering, National Cheng Kung University, Tainan City, Taiwan, 4 Division of Nephrology, Department of Internal Medicine, Changhua Christian Hospital, Changhua, Taiwan, 5 Institute of Clinical Medicine, National Yang-Ming University, Taipei, Taiwan, 6 Department of Medicine, Chung Shan Medical University, Taichung, Taiwan, 7 Department of Hospitality Management, MingDao University, Changhua, Taiwan

* hhchang0210@gmail.com

## Abstract

### Background

Creatinine is widely used to estimate renal function, but this is not practical in critical illness. Low creatinine has been associated with mortality in many clinical settings. However, the associations between predialysis creatinine level, Sepsis-related Organ Failure Assessment (SOFA) score, fluid overload, and mortality in acute kidney injury patients receiving dialysis therapy (AKI-D) has not been fully addressed.

### Methods

We extracted data for AKI-D patients in the eICU and MIMIC databases. We conducted a retrospective observational cohort study using the eICU dataset. The study cohort was divided into the high-creatine group and the low-creatinine group by the median value (4 mg/dL). The baseline patient information included demographic data, laboratory tests, medications, and comorbid conditions. The independent association of creatinine level with 30-day mortality was examined using multivariate logistic regression analysis. In sensitivity analyses, the associations between creatinine, SOFA score, and mortality were analyzed in patients with or without fluid overload. We also carried out an external validity using the MIMIC dataset.

### Results

In all 1,600 eICU participants, the *30*-day mortality rate was 34.2%. The crude overall mortality rate in the low-creatinine group (44.9%) was significantly higher than that in the high-creatinine group (21.9%; *P* < 0.001). In the fully adjusted models, the low-creatinine group was associated with a higher risk of 30-day mortality (odds ratio, 1.77; 95% confidence interval, 1.29–2.42; *P* < 0.001) compared with the high-creatinine group. The low-creatinine group had higher SOFA and nonrenal SOFA scores. In sensitivity analyses, the low-

**Data Availability Statement:** All relevant data are within the paper and its Supporting Information files. Data can also be found here: https://physionet.org/content/mimiciii-demo/1.4/ and https://physionet.org/content/eicu-crd/2.0/.

**Funding:** The author(s) received no specific funding for this work.

**Competing interests:** The authors have declared that no competing interests exist.

creatinine group had a higher 30-day mortality rate with regard to the BMI or albumin level. Fluid overloaded patients were associated with a significantly worse survival in the low-creatinine group. The results were consistent when assessing the external validity using the MIMIC dataset.

## Conclusions

In patients with AKI-D, lower predialysis creatinine was associated with increased mortality risk. Moreover, the mortality rate was substantially higher in patients with lower predialysis creatinine with concomitant elevation of fluid overload status.

## Introduction

Acute kidney injury (AKI) is a common and significant problem in the intensive care unit (ICU), and about 25% of patients with AKI require renal replacement therapy (RRT) [1, 2]. However, a high mortality rate of 30%–50% is noted [3, 4]. Many severity of illness scoring systems have been developed for mortality prediction [5–8]. In terms of the Sepsis-related Organ Failure Assessment (SOFA) score [9], it doesn't have good discrimination in this patient group [5].

Creatinine is a metabolite of creatine and creatine phosphate, which are in the highest concentration in skeletal muscle, and is mainly eliminated via the kidney [10, 11]. Therefore, serum creatinine is used to not only estimate renal function but also to reflect muscle mass. Low serum creatinine is also a marker of malnutrition [11]. However, it is also related to sex, age, diet, and fluid status [12, 13]. Because creatinine is affected by many factors, it usually overestimates renal function in critically ill patients [14]. Studies have shown that low creatinine was associated with high mortality in the ICU and an increased mechanical ventilation use rate, and was also a risk marker of mortality in hemodialysis patients [11, 12, 15]. The reason might relate to fluid overload [13, 16]. Only a few studies with small sample sizes have addressed mortality and the creatinine level in patients with AKI [5, 8, 13, 17].

For this reason, we conducted this retrospective study using two public datasets to explore the associations between predialysis creatinine, SOFA score, fluid overload, and mortality among patients with AKI who were receiving dialysis (AKI-D) in the ICU.

## Methods

### Participants and measurements

This retrospective, observational cohort study was performed using two publicly available ICU datasets, the MIMIC-III [18] and the eICU [19] Collaborative Research Database (eICU-CRD). The MIMIC-III database was released in 2016 by the Massachusetts Institute of Technology Laboratory for Computational Physiology (MIT-LCP) and contained data from a single tertiary care hospital (Beth Israel Deaconess Medical Center). The eICU-CRD is a multicenter critical care database containing data from rural/nonacademic hospitals across the US and was made available in 2018 by Philips Healthcare with the help of researchers from MIT-LCP. There is no overlap in the patients included in these two databases.

We included adult patients 18 years of age or older who received RRT (either intermittent hemodialysis or continuous RRT [CRRT]) in the ICU for AKI. The AKI in this study was defined according to the Kidney Disease Improving Global Outcomes clinical practice guidelines [20] and diagnosis codes. We only used the creatinine criteria because urine data was too

complicated to preprocess in the retrospective databases. For patients who did not have more than one creatinine value to make a comparison, but who had RRT records, we included patients who were diagnosed as having AKI based on their ICD-9 diagnosis codes (S1 Table). If a patient had been admitted to the ICU multiple times in one hospitalization course, data from the ICU admission that included the initial dialysis treatment was extracted for the study. Patients with a history of end-stage kidney disease who underwent chronic peritoneal dialysis or hemodialysis (S1 Table) were excluded from the study. We also excluded patients who had chronic kidney disease (CKD) stage 4 and 5 based on ICD-9 codes (S1 Table), because we were interested in patients who did not have advanced CKD at baseline. Patients with a history of any organ transplant were also excluded as they may have other confounding risk variables that affect mortality. Patients without complete records of vital signs and creatinine data one day before RRT start were excluded.

The variables collected consisted of demographics, medical history, mechanical ventilation use, vital signs, laboratory tests, and medications (diuretics and vasopressors, see S2 Table). The time window of mechanical ventilation, vital signs, laboratory tests, and medications were recorded one day before RRT initiation. Past medical history was extracted from database records using ICD-9 codes (S1 Table). Relevant past medical history included in the study were diabetes mellitus (DM), CKD, hypertension (HTN), congestive heart failure (CHF), liver cirrhosis (LC), and cancer. Vital signs in this study included the mean values of the following variables: shock index (SI), Glasgow Coma Scale (GCS), mean arterial pressure (MAP), respiratory rate (RR), and heart rate (HR). The mean SI was calculated by the formula: SI = mean HR/mean systolic blood pressure. For laboratory tests, we used the mean value of all variables recorded one day before the date of the first dialysis therapy. We excluded the variables with >25% missing values, except for albumin level, because we thought that albumin was an important factor for mortality prediction. S3 Table reveals the percentages of missing data in the laboratory tests. Multiple imputation by chained equations (MICE) with five imputed datasets was used to impute the missing values of the laboratory tests and vital signs and the results were pooled using the MICE package [21].

We modified the codes from https://github.com/nus-mornin-lab/oxygenation_kc and https://github.com/MIT-LCP/mimic-code/tree/master/concepts/severityscores to calculate the SOFA score using variables collected one day before RRT start in the eICU and MIMIC datasets based on methods used in the original study [9]. For patients with missing variables, the SOFA score was imputed using MICE as described previously. We also calculated nonrenal SOFA score excluding the renal component.

The primary aims of the investigation were to assess whether the predialysis creatinine level was associated with 30-day mortality independent of other risk factors and to explore the association between the predialysis creatinine level and the SOFA score.

## Statistical analyses

The study cohort was stratified into two groups according to the median creatinine value. Categorical variables were presented as counts, proportions, and frequencies; continuous variables were expressed as mean with standard deviation. Numeric variables of clinical characteristics between the two groups were compared using the Student's t test. The chi-square test was used to compare the differences of the categorical variables. Logistic regression was used to calculate odds ratios (ORs) and 95% confidence intervals (95% CIs) for the analyses of predictors of mortality. The comparison of survival status between the two groups was done using the Kaplan–Meier curve with significance levels determined by the log rank test. We implemented four models for the adjustments of the covariates: model 1, adjusted for age, sex, and ethnicity;

model 2, adjusted for all variables in model 1 plus DM, HTN, CKD, malignancy, and LC; model 3, adjusted for all variables in model 2 plus GCS, HR, MAP, RR, SI, ICU days before dialysis, diuretics, vasopressors, and mechanical ventilation; model 4, adjusted for all variables in model 3 plus laboratory tests. We used the Kruskal–Wallis test to compare the difference of SOFA and nonrenal SOFA scores between the groups.

Sensitivity analyses were done with ORs and survival curves of creatinine in patients with or without fluid overload. The definition of fluid overload in this study was that (total input amount–total output amount) in liters from ICU admission to RRT initiation > 10% of admission body weight in kilograms. We also performed sensitivity analyses in which we repeated our primary analyses by albumin and BMI (Body Mass Index) level to explore the relationship between creatinine and nutritional status.

To assess external validity, each analysis was repeated using the MIMIC dataset to explore the heterogeneity. Analyses were performed using R version 3.6.1 (R Foundation for Statistical Computing).

## Results

### Baseline characteristics of the study cohort

The cohort from the eICU database included 8,201 patients who required dialysis therapy. Of those patients, 1,600 patients met the inclusion and exclusion criteria for the study. The cohort from the MIMIC database included 3,357 patients who required dialysis therapy. Of those patients, 694 patients met the criteria for inclusion in the study (Fig 1).

The median creatinine level among all 1,600 eICU participants was 3.72 mg/dL. Their mean age was 62.5 ± 14.5 years, and 940 (58.7%) patients were men. The overall mortality rate was 34.2%. The cohort was divided into low- and high-creatinine groups according to the creatinine value (4 mg/dL). All baseline characteristics are summarized in Table 1. Differences in age, gender, laboratory parameters, comorbidities, and laboratory tests between the two groups were statistically significant.

### Creatinine and 30-day mortality

The crude mortality rate was 21.9% (n = 163) for the high-creatinine group and 44.9% (n = 385) for the low-creatinine group ($P < 0.001$). The Kaplan–Meier analysis revealed that the patient survival was significantly worse for the low-creatinine group than for the high-creatinine group ($P < 0.0001$) (Fig 2). The unadjusted and adjusted ORs are presented in Table 2. Compared with the high-creatinine group, the OR for the low-creatinine group was 2.90 (95% CI: 2.33 to 3.62) for 30-day mortality in the unadjusted model. In the fully adjusted model (model 4), the risk of 30-day mortality in the low-creatinine group was 77% higher (OR, 1.77; 95% CI: 1.29 to 2.42).

### Predialysis creatinine and SOFA, nonrenal SOFA scores

Fig 3 shows that the low-creatinine group had higher SOFA and nonrenal SOFA scores than the high-creatinine group ($P < 0.0001$), which reflected the worse survival. The median (interquartile range) values of the SOFA and nonrenal SOFA scores were 12 (10–15) and 10 (8–13), respectively, in the low-creatinine group and 11 (9–14) and 7 (5–10), respectively, in the high-creatinine group.

### Sensitivity analyses

Of 1,600 patients with complete admission BMI and input/output records, 360 (22.5%) patients were fluid overloaded. In the fully adjusted model (model 4 plus BMI, < 35 and ≥ 35

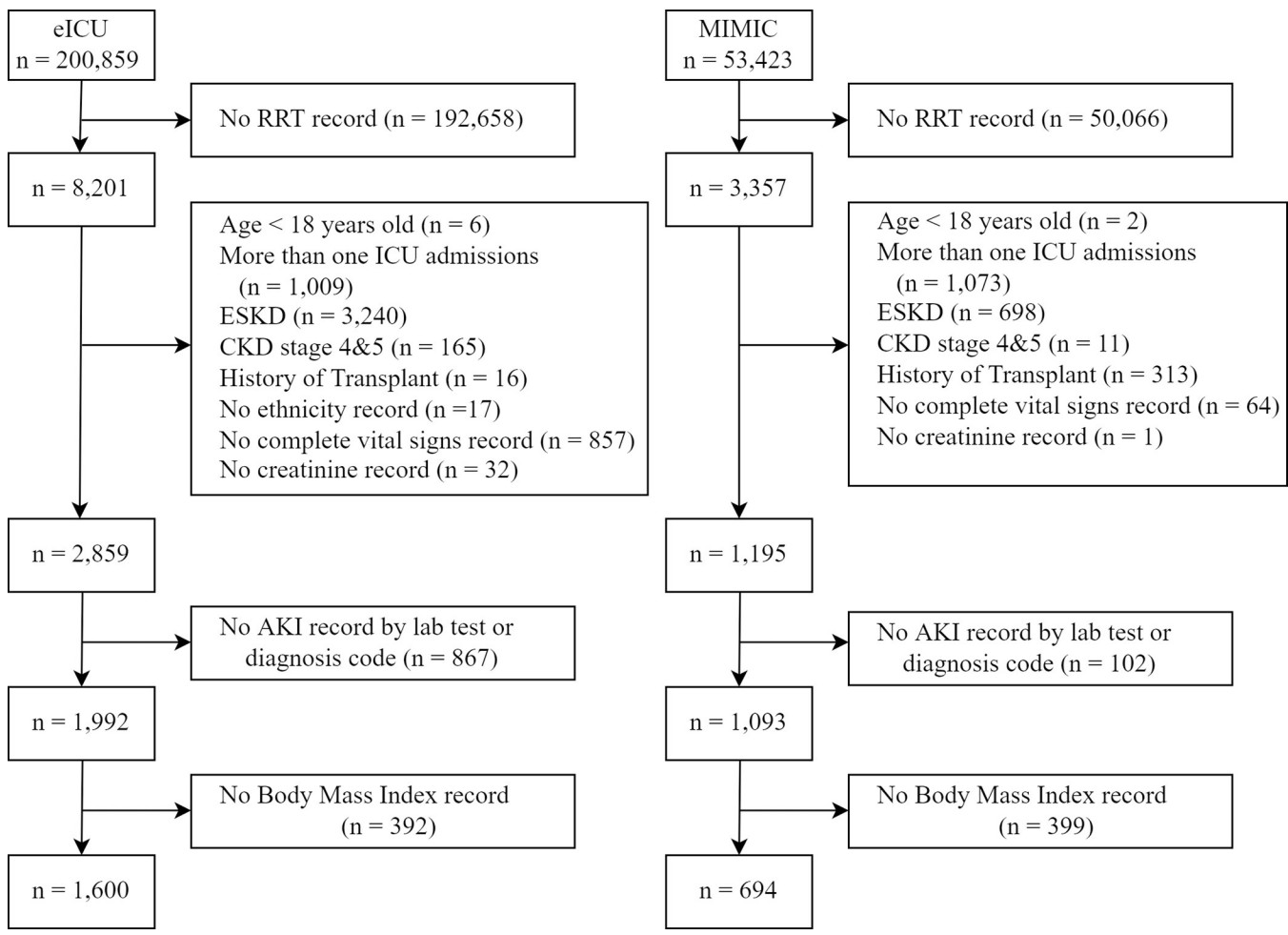

**Fig 1. Participant flow diagram.** n is patient unit encounter. Abbreviations: RRT, renal replacement therapy; ESKD, end-stage kidney disease; CKD, chronic kidney disease; ICU, intensive care unit; AKI, acute kidney injury.

kg/m$^2$), the risk of mortality in the low-creatinine group was 1.77 (95% CI: 1.29 to 2.42) times higher odds of death. There was no significant between BMI groups (OR, 1.13; 95% CI: 0.85–1.51). The unadjusted 30-day mortality rate increased in the fluid-overloaded patients ($P < 0.0001$) (Fig 4A). For all fluid-overloaded patients, the low-creatinine group had a higher 30-day mortality rate ($P < 0.0001$) (Fig 4B) and nonrenal SOFA score ($P < 0.0001$) (Fig 5) than the high-creatinine group. In the low-creatinine group, fluid overloaded patients were associated with a significantly worse survival ($P < 0.001$) (Fig 4C). With regard to the BMI or albumin level, the low-creatinine group still had a higher 30-day mortality rate (S1 Fig).

## External validity

S4 Table reveals the distribution between the eICU and MIMIC datasets. The mortality rate, comorbidity, and many other variables were significantly different. The results of external validity using the MIMIC dataset were consistent with those using the eICU dataset. The low-creatinine group had a higher mortality risk (S5 Table), worse survival (S2 Fig), and higher nonrenal SOFA score (S3 Fig). Of 694 patients with complete admission BMI and input/output records, 264 (38%) patients were fluid overloaded. In fluid-overloaded patients, the

**Table 1. Baseline characteristics of the study population by the median of creatinine level.**

| Variables | Total | Cr <4 mg/dL | Cr ≥4 mg/dL | P value |
|---|---|---|---|---|
| Number of patients | 1,600 | 857 | 743 | |
| Demographics | | | | |
| Age, years | 62.5 ± 14.5 | 63.5 ± 14.5 | 61.5 ± 14.7 | 0.005 |
| Sex, % male | 940 (58.7%) | 448 (52.3%) | 492 (66.2%) | <0.001 |
| Black race, % | 199 (12.4%) | 77 (9.0%) | 122 (16.4%) | <0.001 |
| Comorbidity, % | | | | |
| DM | 180 (11.2%) | 82 (9.6%) | 98 (13.2%) | 0.027 |
| Hypertension | 179 (11.1%) | 81 (9.5%) | 98 (13.2%) | 0.022 |
| CHF | 240 (15.0%) | 132 (15.4%) | 108 (14.5%) | 0.679 |
| CKD | 206 (12.8%) | 91 (10.6%) | 115 (15.5%) | 0.005 |
| Malignancy | 76 (4.7%) | 53 (6.2%) | 23 (3.1%) | 0.005 |
| Liver Cirrhosis | 102 (.36%) | 70 (8.2%) | 32 (4.3%) | 0.002 |
| Medication, % | | | | |
| Diuretics | 197 (12.3%) | 127 (14.8%) | 70 (9.4%) | 0.001 |
| Vasopressors | 699 (43.6%) | 466 (54.4%) | 233 (31.4%) | <0.001 |
| Laboratory data | | | | |
| BUN (mg/dL) | 62.0 ± 37.1 | 47.7 ± 29.2 | 78.6 ± 38.5 | <0.001 |
| FiO$_2$ (%) | 51.2 ± 25.8 | 55.0 ± 24.9 | 47.1 ± 26.3 | <0.001 |
| Hgb (mg/dL) | 9.7 ± 2.1 | 9.6 ± 2.1 | 9.9 ± 2.0 | 0.004 |
| O$_2$ Sat (%) | 95.1 ± 5.8 | 94.8 ± 6.0 | 95.4 ± 5.6 | 0.039 |
| WBC(×10$^3$/μL) | 15.9 ± 24.1 | 17.0 ± 31.7 | 14.8 ± 9.5 | 0.070 |
| Albumin (g/dL) | 2.6 ± 0.7 | 2.6 ± 0.7 | 2.7 ± 0.7 | <0.001 |
| HCO3 (mmol/L) | 20.4 ± 5.7 | 21.2 ± 5.7 | 19.6 ± 5.7 | <0.001 |
| AG (mmol/L) | 14.6 ± 6.3 | 13.3 ± 5.8 | 16.2 ± 6.6 | <0.001 |
| Calcium(mg/dL) | 8.0 ± 1.0 | 8.0 ± 1.0 | 8.0 ± 1.1 | 0.464 |
| Glucose (mg/dL) | 149.9 ±70.7 | 150.0 ± 59.2 | 149.8 ± 82 | 0.935 |
| Platelet(×10$^3$/μL) | 179.2±111.9 | 155.7 ± 104 | 206.3±114.7 | <0.001 |
| K (mmol/L) | 4.6 ± 1.0 | 4.4 ± 0.9 | 4.9 ± 1.1 | <0.001 |
| Na (mmol/L) | 137.9 ± 6.3 | 139.2 ± 5.9 | 136.6 ± 6.6 | <0.001 |
| GCS score | 10.8 ± 3.8 | 10.0 ± 3.8 | 11.7 ± 3.8 | <0.001 |
| HR (BPM) | 89.6 ± 18.0 | 91.6 ± 18.5 | 87.3 ± 17.2 | <0.001 |
| MAP (mmHg) | 76.1 ± 14.4 | 75.0 ± 12.9 | 77.5 ± 15.9 | <0.001 |
| RR (BPM) | 21.1 ± 5.5 | 21.6 ± 5.5 | 20.5 ± 5.5 | <0.001 |
| SI | 0.8 ± 0.2 | 0.8 ± 0.2 | 0.8 ± 0.2 | <0.001 |
| Days of ICU stay before RRT initiation | 2.8 ± 4.2 | 3.5 ± 4.7 | 2.1 ± 3.3 | <0.001 |
| CRRT, % | 466 (29.2%) | 336 (39.2%) | 130 (17.5%) | <0.001 |
| Death, % | 548 (34.2%) | 385 (44.9%) | 163 (21.9%) | <0.001 |
| MV, % | 1251(78.2%) | 756 (88.2%) | 495 (66.6%) | <0.001 |
| Fluid overload | 360 (22.5) | 258 (30.1%) | 102 (13.7) | <0.001 |
| BMI | 31.7 ± 9.9 | 31.2 ± 9.5 | 32.5 ± 10.4 | 0.010 |

Data are presented as mean ± standard deviation for continuous variables and number (%) for categorical variables.

*Abbreviations*: CHF, congestive heart failure; CKD, chronic kidney disease; BUN, blood urea nitrogen; FiO$_2$, fraction of inspired oxygen; Hgb, hemoglobin; WBC, white blood cell; GCS, Glasgow Coma Scale; HR, heart rate; MAP, mean arterial pressure; RR, respiratory rate; SI, shock index; ICU, intensive care unit; RRT, renal replacement therapy; CRRT, continuous renal replacement therapy; MV, mechanical ventilation.

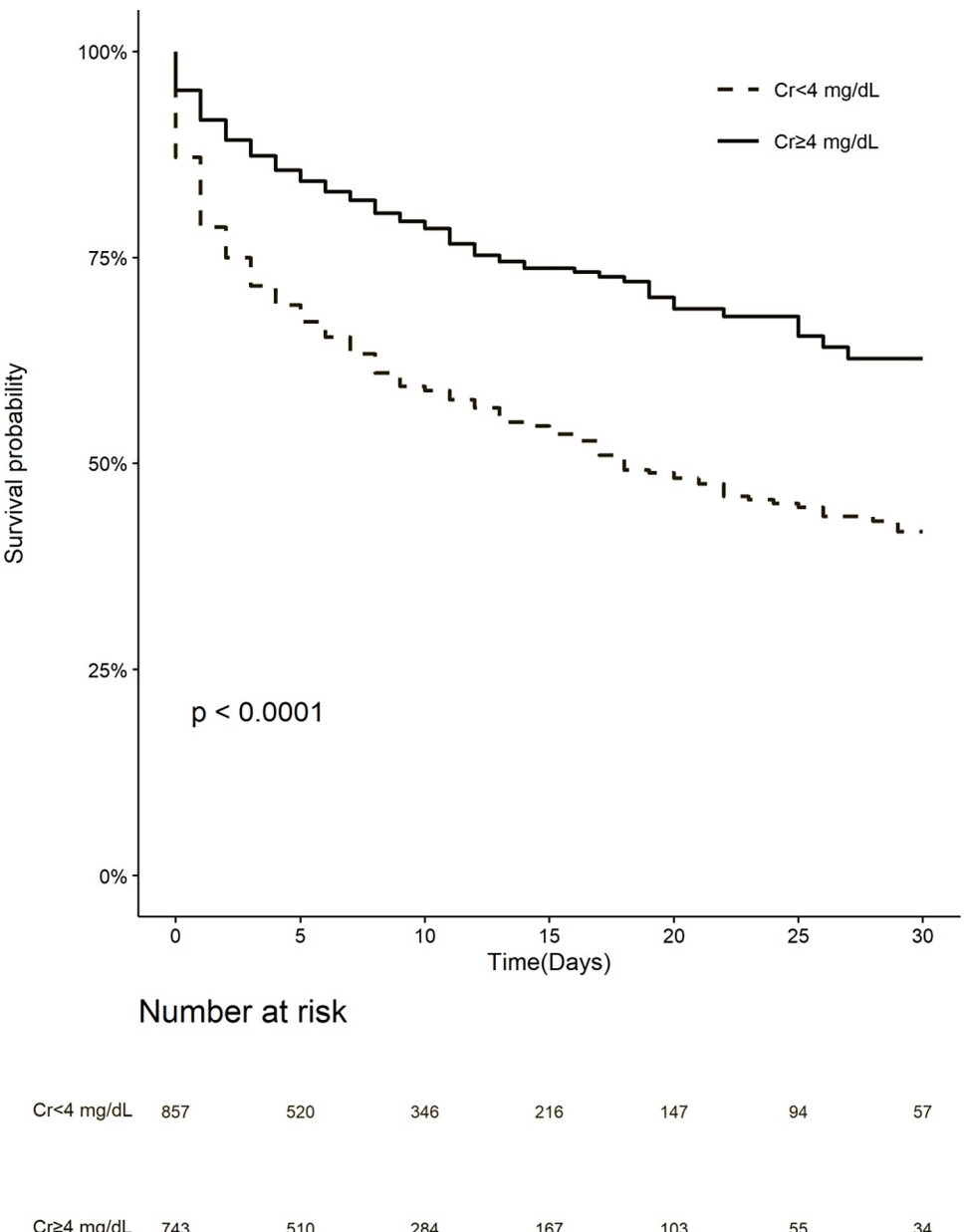

**Fig 2. Kaplan–Meier curve of mortality according to creatinine category.** The low-creatinine (Cr < 4 mg/dL) group was associated with worse survival than the high-creatinine (Cr ≥ 4 mg/dL) group.

mortality risk in the low-creatinine group was 2.67 times higher (OR, 2.67; 95% CI: 1.24 to 5.94, $P = 0.01$) in the fully adjusted model (model 4 plus BMI). The low-creatinine group also had a higher 30-day mortality rate (S4 Fig) and nonrenal SOFA score (S5 Fig).

## Discussion

In this study of AKI-D patients, we identified that a low creatinine level was independently associated with 30-day mortality. The low-creatinine (<4 mg/dL) group was associated with a 77% higher risk of mortality in these patients. The SOFA and nonrenal SOFA scores in the low-creatinine group were higher than those in the high-creatinine group, indicating that

**Table 2. Risk of mortality in low predialysis creatinine patients compared with high predialysis creatinine patients.**

| Significant variables | Unadjusted OR (95% CI) | Model 1 | Model 2 | Model 3 | Model 4 |
|---|---|---|---|---|---|
| Creatinine < 4 mg/dL | 2.90 (2.33–3.62) | 2.82 (2.26–3.54) | 2.69 (2.14–3.39) | 1.67 (1.28–2.19) | 1.77 (1.29–2.42) |
| Age | 1.02 (1.01–1.03) | 1.02 (1.01–1.03) | 1.02 (1.02–1.03) | 1.04 (1.03–1.05) | 1.04 (1.03–1.06) |
| Liver cirrhosis | 2.60 (1.73–3.91) | | 2.77 (1.81–4.28) | 2.31 (1.42–3.78) | 2.00 (1.20–3.36) |
| Vasopressor | 3.60 (2.91–4.48) | | | 1.55 (1.18–2.04) | 1.48 (1.11–1.98) |
| Glasgow Coma Scale | 0.82 (0.80–0.85) | | | 0.82 (0.80–0.85) | 0.88 (0.84–0.91) |
| Mean arterial pressure | 0.96 (0.95–0.97) | | | 0.96 (0.95–0.97) | 0.99 (0.97–1.00) |
| Shock index | 13.67 (8.48–22.30) | | | 13.6 (8.48–22.30) | 3.42 (0.96–12.90) |
| CRRT | 3.12 (2.49–3.91) | | | 1.64 (1.25–2.15) | 1.48 (1.12–1.97) |
| Respiratory rate | 1.09 (1.07–1.11) | | | 1.09 (1.07–1.11) | 1.04 (1.02–1.07) |
| Albumin | 0.69 (0.59–0.80) | | | | 0.88 (0.72–1.08) |
| Anion gap | 1.03 (1.02–1.05) | | | | 1.05 (1.02–1.08) |
| Calcium | 0.84 (0.75–0.93) | | | | 1.10 (0.95–1.26) |
| Platelet | 1.00 (0.99–1.00) | | | | 1.00 (0.99–1.00) |
| FiO$_2$ | 1.02 (1.02–1.03) | | | | 1.01 (1.01–1.02) |

The referent group for all models is creatinine above the median of 4 mg/dL. The variables for adjustments in models 1–4 are described.

Model 1: creatinine, age, sex, and ethnicity. Model 2: model 1 plus diabetes mellitus, hypertension, chronic kidney disease, malignancy, and liver cirrhosis. Model 3: model 2 plus Glasgow Coma Scale, heart rate, mean arterial pressure, respiratory rate, shock index, days in the ICU before dialysis, CRRT, diuretics, vasopressors, and mechanical ventilation. Model 4: model 3 plus blood urea nitrogen, FiO$_2$, hemoglobin, white blood cell count, albumin, HCO$_3$, anion gap, calcium, glucose, platelet, potassium, and sodium.

*Abbreviations*: OR, odds ratio; 95% CI, 95% confidence interval; ICU, intensive care unit; CRRT, continuous renal replacement therapy

AKI-D patients with low predialysis creatinine values have more organ dysfunctions. Furthermore, patients with lower predialysis creatinine had a significantly higher mortality rate as the degree of fluid overload. The results were consistent assessing external validity in the MIMIC database.

A low creatinine level can relate to an increase in excretion or a decrease in generation. Real kidney function improvement was uncommon in critically ill patients [22], whereas those patients usually have more complicated underlying problems that affect creatinine generation, such as sepsis [23], poor nutrition status and low muscle mass [11, 24], liver failure, and older age [24]. The relationship between low creatinine at the start of RRT and mortality in AKI-D patients was demonstrated in previous reports [5, 8]. Our results showed that the low-creatinine group had a higher mean age, a lower mean albumin value, a elevated SOFA and nonrenal SOFA scores, and a higher proportion of patients with LC, as well as increased mortality. Besides, fluid overload in AKI patients was an independent factor for mortality and would lead to low creatinine [25, 26]. Fluid overload can result in tissue edema and organ dysfunctions and increased risk of mortality [13, 25, 27, 28]. In this study, we observed patients with fluid overload had a higher mortality rate, especially when they also had lower creatinine levels.

In addition to causing organ dysfunctions, fluid overload is also the result of organ failure. In other words, there may be other causes than fluid overload in relation to low creatinine associated with high mortality. The hypothetical reason is that the more severely and rapidly critical illnesses develop, the lower predialysis creatinine is in AKI-D patients. Creatinine needs time to achieve a steady state in AKI patients [29]. However, for severe and critically ill patients, there would not be enough time for the creatinine to reach a steady state when RRT starts. In the present study, patients with fluid overload, the low-creatinine group had higher SOFA and nonrenal SOFA scores. Therefore, AKI-D patients with low predialysis creatinine

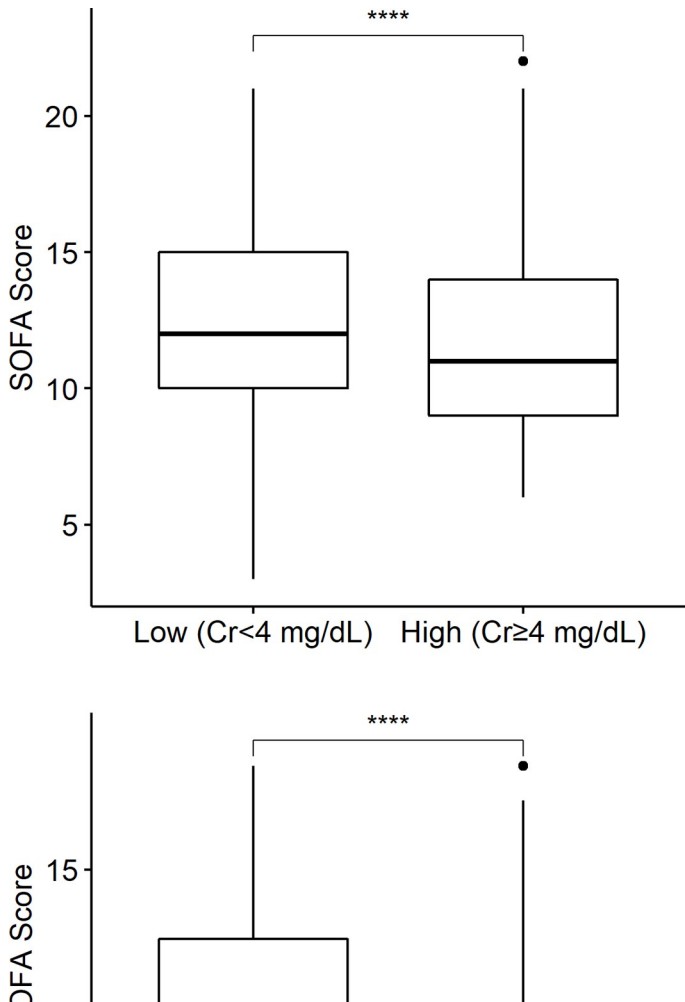

**Fig 3. Box plots of SOFA and nonrenal SOFA scores between the low-creatinine (Cr < 4 mg/dL) group and with high-creatinine (Cr ≥ 4 mg/dL) group.** The low-creatinine (Cr < 4 mg/dL) group had higher SOFA and nonrenal SOFA scores (*Kruskal*–Wallis test, *P* < 0.0001).

implicitly have more organ dysfunctions, which reflects worse survival. Based on our study, the impact of creatinine should be considered in scoring systems used for AKI-D patients, as with the Acute Physiology And Chronic Health Evaluation (APACHE) score [30], the HEpatic failure, LactatE, NorepInephrine, medical Condition, and Creatinine (HELENICC) score [8], and the ATN score [5].

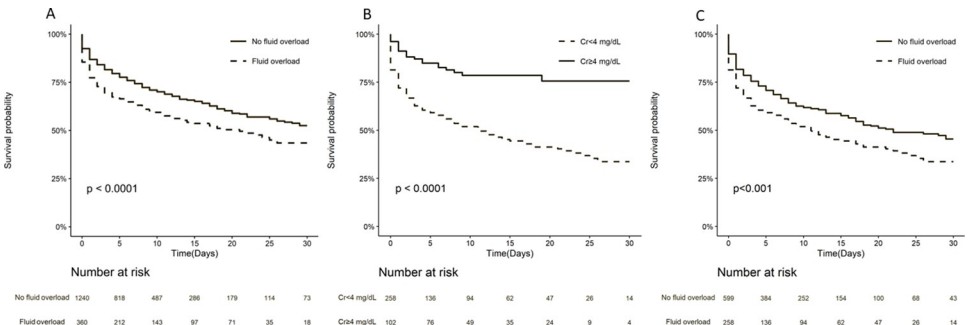

**Fig 4. Kaplan–Meier survival curves for 30-day mortality in patients with or without fluid overload in the eICU dataset.** A. The unadjusted 30-day mortality rate increased in the fluid-overloaded group. B. In patients with fluid overload, the low-creatinine (Cr < 4 mg/dL) group was associated with worse survival. C. In low-creatinine group, fluid-overloaded patients were associated with worse survival.

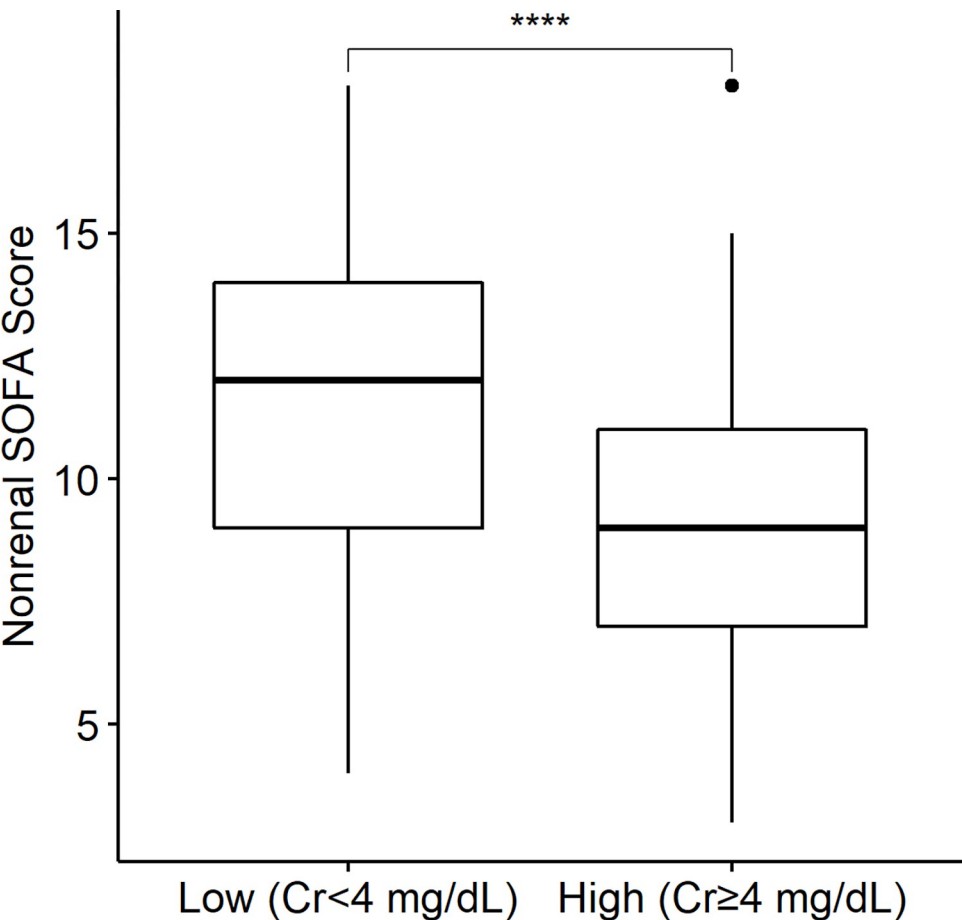

**Fig 5. Box plot of nonrenal SOFA score between the low-creatinine (Cr < 4 mg/dL) group and the high-creatinine (Cr ≥ 4 mg/dL) group in fluid-overloaded patients.** The low-creatinine (Cr < 4 mg/dL) group had a higher nonrenal SOFA score (*Kruskal*–Wallis test, $P < 0.0001$).

## Strengths and limitations

This study was based on the eICU database, which has a large sample size and is from distinct regions and hospitals across the US. Moreover, the findings were validated via sensitivity analyses and external validity and were consistent in these two datasets. However, there were several limitations to our study. First, as previously mentioned, some data were missing and the urine output data was difficult to preprocess in the retrospective databases. Second, the etiologies of AKI and indications of RRT may be helpful for further analysis, but they were not mentioned in these two databases. Third, the baseline creatinine before ICU admission was not recorded so that we can not notice the change of creatinine from baseline to RRT start. Finally, the cause of death was not recorded, though knowing causes of death is important for clinical practice.

## Conclusion

In patients with AKI-D, the low-creatinine group had a significantly higher risk of mortality compared with that of the high-creatinine group. Moreover, the mortality rate was substantially higher in patients with lower predialysis creatinine with concomitant elevation of fluid overload status.

## Supporting information

**S1 Table. ICD-9 diagnosis codes used to identify acute kidney injury, transplant history, and comorbidities.**
(DOCX)

**S2 Table. Drug names of diuretics and vasopressors.**
(DOCX)

**S3 Table. Percentages of missing data.**
(DOCX)

**S4 Table. Comparison of variables between the eICU and MIMIC datasets.**
(DOCX)

**S5 Table. Risk of mortality in patients with high predialysis creatinine levels compared with patients with low predialysis creatinine levels in the MIMIC dataset.**
(DOCX)

**S1 Fig. Kaplan–Meier survival curves for 30-day mortality by albumin and BMI in the eICU dataset.** The low-creatinine group had a higher 30-day mortality rate in four groups.
(PNG)

**S2 Fig. Kaplan–Meier survival curve for 30-day mortality according to creatinine category in the MIMIC dataset.** The low-creatinine (Cr < 4 mg/dL) group was associated with worse survival (log rank test, $P < 0.0001$).
(PNG)

**S3 Fig. Box plots of SOFA and nonrenal SOFA scores between patients with low creatinine (Cr < 4 mg/dL) and with high creatinine (Cr ≥ 4 mg/dL) in the MIMIC dataset.** The low-creatinine (Cr < 4 mg/dL) group had a higher nonrenal SOFA score (*Kruskal*–Wallis test, $P < 0.0001$). ns: not significant.
(PNG)

**S4 Fig. Kaplan–Meier survival curve for 30-day mortality according to creatinine category in fluid-overloaded patients in the MIMIC dataset.** The low-creatinine (Cr < 4 mg/dL) group was associated with worse survival (log rank test, $P < 0.001$).
(PNG)

**S5 Fig. Box plot of nonrenal SOFA score between fluid-overloaded patients with low creatinine (Cr < 4 mg/dL) and with high creatinine (Cr ≥ 4 mg/dL) in the MIMIC dataset.** The low-creatinine (Cr < 4 mg/dL) group had a higher nonrenal SOFA score (*Kruskal*–Wallis test, $P < 0.001$).
(PNG)

## Author Contributions

**Conceptualization:** Hsin-Hsiung Chang, Chia-Lin Wu.

**Data curation:** Hsin-Hsiung Chang, Chun-Chieh Tsai.

**Formal analysis:** Chun-Chieh Tsai.

**Methodology:** Hsin-Hsiung Chang, Ping-Fang Chiu.

**Writing – original draft:** Hsin-Hsiung Chang.

**Writing – review & editing:** Hsin-Hsiung Chang.

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
