## [Decision Letter · Decision Letter 0]

17 Feb 2022

PONE-D-21-22096

Association between predialysis creatinine and mortality in acute kidney injury patients requiring dialysis

PLOS ONE

Dear Dr. Chang,

Thank you for submitting your manuscript to PLOS ONE. After careful consideration, we feel that it has merit but does not fully meet PLOS ONE’s publication criteria as it currently stands. Therefore, we invite you to submit a revised version of the manuscript that addresses the points raised during the review process.

The manuscript focuses on a topic of potential interest. The study, however, has some shortcomings that should be addressed. In particular,  to mention  some of them, i) need to provide additional nutrition parameters (i.e. BMI), if available, and include them on the analysis; ii) need to provide in the survival analysis (Kaplan-Meier curves) in Figure 2A, the evaluation by albumin level and, if available, by BMI; iii) need to add also in Figure 2B albumin and BMI, and perform further analyses; iv) need to clarify in both the abstract (Results section) and in the Results section itself (page 12), the sentence about the sensitivity analysis which showed that low creatinine patients with fluid overload had worse survival; v) please comment the issue that dichotomising patients according to serum creatinine is relatively simplistic way to evaluate renal function given that creatinine can be insensitive measure of renal function in the critically ill, as many dynamic changes occur concomitantly in the acute setting that can affect this; vi) need to clarify on page 11 whether there is a statistically significant difference in SOFA/non-renal SOFA scores between the low and high creatinine groups, and add the relevant P-value to this section.

We look forward to receiving your revised manuscript.

Kind regards,

Giuseppe Remuzzi

Academic Editor

PLOS ONE

Journal Requirements:

2. Please include your tables as part of your main manuscript and remove the individual files. Please note that supplementary tables (should remain/ be uploaded) as separate "supporting information" files"

Reviewers' comments:

Reviewer's Responses to Questions

**Comments to the Author**

1. Is the manuscript technically sound, and do the data support the conclusions?

Reviewer #1: Yes

Reviewer #2: Yes

2. Has the statistical analysis been performed appropriately and rigorously? 

Reviewer #1: Yes

Reviewer #2: Yes

3. Have the authors made all data underlying the findings in their manuscript fully available?

Reviewer #1: Yes

Reviewer #2: Yes

4. Is the manuscript presented in an intelligible fashion and written in standard English?

Reviewer #1: Yes

Reviewer #2: Yes

5. Review Comments to the Author

Reviewer #1: 1.Some review of the way it is written (background) that can be easily corrected by the authors.

2.The authors presented evidence of low creatinine as a risk factor for mortality, if possible to extract from the database more nutritional parameters need to be included i.e BMI .Also if in the table 2 BMI can be included because the low creatinine may reflect a compromised nutritional status (reason for which BMI and other nutritional factors could be included).

3.In the survival analysis (Kaplan Meir curves) in figure 2A analysis by albumin level can be done and may give some answers also BMI can be used if available.Figure2B is it possible to add albumin and BMI and analyze and see if similar results are obtained.

Reviewer #2: This is a well-written paper on an important topic using two large international ICU datasets (eICU and MIMIC) for analysis followed by external validation using a separate cohort that differed significantly in many ways from the original study population; increasing the potential generalisability of the results. I found the introduction, discussion, statistical methodology and presentation of results to be robust and thought-provoking with regard to generation of further hypotheses in this field. One criticism is that in both the Abstract (Results section) and the Results section itself (page 12), the sentence about the sensitivity analysis which showed that low creatinine patients with fluid overload had worse survival is very poorly and unclearly phrased: "In sensitivity analyses, the low-creatinine group was associated with higher mortality rate in patients as the degree of fluid overload". Please make this phrasing more clear e.g. "In the low-creatinine group, fluid overloaded patients were associated with a significantly worse survival". In terms of the research methodology, dichotomising patients according to serum creatinine is a relatively simplistic way to evaluate renal function, given that creatinine can be an insensitive measure of renal function in the critically ill as many dynamic changes occur concomitantly in the acute setting that can affect this. Nevertheless, the methodology used and results presented are interesting and worthy of attention; particularly given that they were generated from and validated in very large ICU population epidemiological datasets. In the Results section, I note that the "Predialysis creatinine and SOFA/nonrenal SOFA scores" section on page 11 does not clarify whether there is a statistically significant difference in SOFA/non-renal SOFA scores between the low and high creatinine groups. Please add the relevant P-value to this section, as per Figure 3. The discussion is well-written and nuanced with regard to the reasons why low creatinine may be associated with these outcomes in critically ill patients; as well as appropriately acknowledging the relevant limitations to this retrospective observational study, albeit large and robustly conducted. It is particularly important to note that the precise cause of death was not routinely documented in these datasets; which may in some cases affect the degree to which it can be reasonably associated with the low creatinine status of the patient. Overall, however, I think this is a very well-constructed and thought-provoking paper that adds important information to the field of AKI outcomes in the critically ill.

6. PLOS authors have the option to publish the peer review history of their article (what does this mean?). If published, this will include your full peer review and any attached files.

Reviewer #1: **Yes: **Luis A Concepcion

Reviewer #2: No

---

## [Author Response · Author response to Decision Letter 0]

11 Apr 2022

We would like to thank the editors and reviewers for the comprehensive assessments, constructive criticisms and valuable comments to our manuscript. Under the recommendations, we have revised our manuscript in response to all comments of one editor and two reviewers. We truly believe that our revised manuscript becomes more clarified with satisfactory changes. The major changes are listed below:

1. The style has been revised to fit all requirements of PLOS ONE guideline (requested by the Editor).

2. We have included our tables as part of our main manuscript and made (requested by the Editor).

3. We removed the reference #10 Kang MW, Kim J, Kim DK, Oh K-H, Joo KW, Kim YS, et al. Machine learning algorithm to predict mortality in patients undergoing continuous renal replacement therapy. Critical Care. 2020;24(1):42. because it focused on the prediction of mortality by machine learning.

4. We added the new affiliation to the corresponding author because he got the new offer in the pandemic and had a lot of supports to revised the manuscript in this new hospital.

5. We have clarified most of all questions requested by the reviewer #1.

6. We have clarified all questions requested by the reviewer #2.

---

## [Decision Letter · Decision Letter 1]

10 Jun 2022

PONE-D-21-22096R1Association between predialysis creatinine and mortality in acute kidney injury patients requiring dialysisPLOS ONE

Dear Dr. Chang,

Thank you for submitting your manuscript to PLOS ONE. After careful consideration, we feel that it has merit but does not fully meet PLOS ONE’s publication criteria as it currently stands. Therefore, we invite you to submit a revised version of the manuscript that addresses the points raised during the review process. The revised manuscript is improved. However, few minor points remain to be addressed. In particular, i) suggestion for including the results of BMI in table 1; ii) clarify how the BMI groups were separated to do the analysis of survival; iii) clarify what percentage of patients were fluid overload in each group (low/high creatinine) and include the number in table 1; iv) need to mention in page 23 that in the group with a creatinine level >4 there were more male and black race patients that usually have higher baseline creatinine levels; v) need to mention that the patients with lower creatinine have more days in the ICU before the RRT initiation; vi) please comment that the creatinine level in these patients may not be the “real” one due to fluid overload and dilution. Did the authors consider to adjust creatinine values to compensate for fluid overload? vi) it may be worth adding a sentence to the limitation section about the limitation associated with dichotomizing creatinine as a continuous predictor variable.

We look forward to receiving your revised manuscript.

Kind regards,

Giuseppe Remuzzi

Academic Editor

PLOS ONE

Journal Requirements:

Reviewers' comments:

Reviewer's Responses to Questions

**Comments to the Author**

1. If the authors have adequately addressed your comments raised in a previous round of review and you feel that this manuscript is now acceptable for publication, you may indicate that here to bypass the “Comments to the Author” section, enter your conflict of interest statement in the “Confidential to Editor” section, and submit your "Accept" recommendation.

Reviewer #1: All comments have been addressed

Reviewer #2: All comments have been addressed

2. Is the manuscript technically sound, and do the data support the conclusions?

Reviewer #1: Yes

Reviewer #2: Yes

3. Has the statistical analysis been performed appropriately and rigorously? 

Reviewer #1: Yes

Reviewer #2: Yes

4. Have the authors made all data underlying the findings in their manuscript fully available?

Reviewer #1: Yes

Reviewer #2: Yes

5. Is the manuscript presented in an intelligible fashion and written in standard English?

Reviewer #1: Yes

Reviewer #2: Yes

6. Review Comments to the Author

Reviewer #1: Review comments to the author:

1.Minor corrections to consider,in page 2 :”creatinine is not practical” probably will chose a different word i.e reliable to reflect the renal function.It is practical because it can be done easily even at the bedside.

2.Minor correction to consider page 3:”low creatinine group had a higher 30 day mortality with regard to the BMI or albumin level” do you mean independent of the BMI or albumin level?.

Same page in conclusion:”mortality rate was substantially higher in patients with lower predialysis creatinine with concomitant elevation of fluid overload” ,do you mean lower predialysis creatinine and fluid overload?.

3.Page 12:we requested analysis using BMI.Is it possible to include the results of the BMI in table 1

What is the BMI between the groups studies (low or high creatinine level)?

How do you separate the BMI groups to do the analysis of survival? By the median distribution?.or using the criteria of low BMI as a reflection of malnutrition i.e BMI<18.5 that indicates underweight?

Same applies for fluid overload:

What percentage of patient were fluid overload in each group (low/high creatinine) include the numbers in table 1 for clarity.

4.page 23:

Consider mention that in the group with a creatinine level >4 there were more males and black race patients that usually have higher baseline creatinine levels.

Also important to mention that the patients with lower creatinine have more days in the ICU before the RRT initiation this could also reflect that they receive more fluid (that is why is important to include the % of patients in this group that were fluid overloaded).

Final comment is that the creatinine level in this patients may not be the “real” one due to fluid overload and dilution (Macedo et al Critical Care 14,R82 (2010) in this paper they have a adjusted creatinine to compensate for fluid overload.

Adjusted creatinine= serum creatinine x( 1+ cumulative fluid balance in L/admission weight (kg) x 0.6)

Will the authors consider using this in their paper?.

Reviewer #2: Thank you for addressing the issues that were previously raised in the original manuscript draft. This revision reads very well and is much more clear. Again it may be worth adding a sentence to the limitations section about the limitation associated with dichotomising creatinine as a continuous predictor variable and I would again ensure that your phrasing is clear in the sentence which addresses mortality being higher in fluid overloaded patients with low pre-dialysis creatinine, as this remains a little unclear in how it is phrased in the results section. Otherwise this is a very well-written revision of a well-conducted, statistically robust and interesting paper.

7. PLOS authors have the option to publish the peer review history of their article (what does this mean?). If published, this will include your full peer review and any attached files.

Reviewer #1: **Yes: **Luis A Concepcion

Reviewer #2: No

---

## [Author Response · Author response to Decision Letter 1]

19 Jun 2022

Response to the Editors’ and Reviewers’ Comments

We would like to thank the editors and reviewers for the comprehensive assessments, constructive criticisms and valuable comments to our manuscript. Under the recommendations, we have revised our manuscript in response to all comments of one editor and two reviewers. We truly believe that our revised manuscript becomes more clarified with satisfactory changes.

Response to Reviewer: 

1. suggestion for including the results of BMI in table 1; clarify how the BMI groups were separated to do the analysis of survival; iii) clarify what percentage of patients were fluid overload in each group (low/high creatinine) and include the number in table 1.

Response:

We appreciate the reviewer’s critical comments and valuable suggestions. We added BMI and albumin to explore the relationship between creatinine and nutritional status. But we chose to put the results in sensitivity analysis section and put them in supplementary files (S5 figure) because there are fewer cases if we include BMI and fluid status. We described the percentage of patients with fluid overload in sensitivity analysis section. We didn’t put the data in table1.

2. Need to mention in page 23 that in the group with a creatinine level >4 there were more male and black race patients that usually have higher baseline creatinine levels; need to mention that the patients with lower creatinine have more days in the ICU before the RRT initiation.

Response:

We thank the reviewer for the excellent comment. We added the comments in the Baseline Characteristics of the Study Cohort section.

3. Please comment that the creatinine level in these patients may not be the “real” one due to fluid overload and dilution. Did the authors consider to adjust creatinine values to compensate for fluid overload? 

Response:

We thank the reviewer for the comment. The fluid status may affect creatinine level. But, a study from Finland recently showed that adjusting Cr values for estimated fluid balance probably has limited value for improving AKI prognostic value for important ICU outcomes (Acta Anaesthesiol Scand 2021 Sep;65(8):1079-1086). We tried to adjust Cr according to Macedo (Critical Care. 2010;14:R82.) and odds ratio of the high-creatinine group was 0.6 (P=0.001). The low-creatinine group still had a higher mortality.

4. It may be worth adding a sentence to the limitation section about the limitation associated with dichotomizing creatinine as a continuous predictor variable.

Response:

We thank the reviewer for the comment. We added it to the Strengths and Limitations section.

---

## [Decision Letter · Decision Letter 2]

1 Aug 2022

PONE-D-21-22096R2Association between predialysis creatinine and mortality in acute kidney injury patients requiring dialysisPLOS ONE

Dear Dr. Chang,

Thank you for submitting your manuscript to PLOS ONE. After careful consideration, we feel that it has merit but does not fully meet PLOS ONE’s publication criteria as it currently stands. Therefore, we invite you to submit a revised version of the manuscript that addresses the points raised during the review process.

The re-revised manuscript is further improved. However, minor points are still pending to be addressed, namely, i) BMI data important to be presented for each group in Table 1; ii) need to be presented in Table 1 also the percentage of patients with fluid overload in each group.

We look forward to receiving your revised manuscript.

Kind regards,

Giuseppe Remuzzi

Academic Editor

PLOS ONE

Journal Requirements:

Reviewers' comments:

Reviewer's Responses to Questions

**Comments to the Author**

1. If the authors have adequately addressed your comments raised in a previous round of review and you feel that this manuscript is now acceptable for publication, you may indicate that here to bypass the “Comments to the Author” section, enter your conflict of interest statement in the “Confidential to Editor” section, and submit your "Accept" recommendation.

Reviewer #1: All comments have been addressed

2. Is the manuscript technically sound, and do the data support the conclusions?

Reviewer #1: Yes

3. Has the statistical analysis been performed appropriately and rigorously? 

Reviewer #1: Yes

4. Have the authors made all data underlying the findings in their manuscript fully available?

Reviewer #1: Yes

5. Is the manuscript presented in an intelligible fashion and written in standard English?

Reviewer #1: Yes

6. Review Comments to the Author

Reviewer #1: I reviewed the authors response.

Still believe that BMI data will be important to be presented for each group in table 1.

Also the percentage of patients with fluid overload in each group to be presented in table 1.

Both data will enhance the differences between the groups.

7. PLOS authors have the option to publish the peer review history of their article (what does this mean?). If published, this will include your full peer review and any attached files.

Reviewer #1: **Yes: **Luis A Concepcion

---

## [Author Response · Author response to Decision Letter 2]

20 Aug 2022

We would like to thank the editors and reviewers for the comprehensive assessments, constructive criticisms and valuable comments to our manuscript. Under the recommendations, we have revised our manuscript in response to all comments of one editor and two reviewers. We truly believe that our revised manuscript becomes more clarified with satisfactory changes. The major changes are listed below:

1. We appreciate the valuable comments. We revised the manuscript as comments. We included BMI and fluid overload variables in our study. There were 1600 and 694 patients in the eICU and MIMIC, respectively. We revised all the tables and figures accoring to the new study group and also the statitistic results.

---

## [Editor Report · Decision Letter 3]

7 Sep 2022

Association between predialysis creatinine and mortality in acute kidney injury patients requiring dialysis

PONE-D-21-22096R3

Dear Dr. Chang,

We’re pleased to inform you that your manuscript has been judged scientifically suitable for publication and will be formally accepted for publication once it meets all outstanding technical requirements.

**The new version of the manuscript is definitely improved. The authors have now adequately addressed the few remaining issues raised by the reviewers.**

Kind regards,

Giuseppe Remuzzi

Academic Editor

PLOS ONE
---

## [Editor Report · Acceptance letter]

16 Sep 2022

PONE-D-21-22096R3 

Association between predialysis creatinine and mortality in acute kidney injury patients requiring dialysis 

Dear Dr. Chang:

I'm pleased to inform you that your manuscript has been deemed suitable for publication in PLOS ONE. Congratulations! Your manuscript is now with our production department. 

Kind regards, 

on behalf of

Prof. Giuseppe Remuzzi 

Academic Editor

PLOS ONE